# Antibiofilm efficacy of 1,064-nm neodymium-doped yttrium aluminum garnet laser on *Trichophyton rubrum* and *Candida albicans*: *in vitro* study

Ji-In Seo,[1] Jin-Woo Lee,[2] Min Kyung Shin[1]

**ABSTRACT**   Fungal biofilms are the most important virulence factors of onychomycosis. However, traditional antifungal therapies lack antibiofilm efficacy. Although neodymium-doped yttrium aluminum garnet (Nd:YAG) laser is one of the most common laser therapies used for onychomycosis, its usage is restricted to aesthetic endpoints. Therefore, this study aimed to assess the antibiofilm activity of Nd:YAG against *Trichophyton rubrum* (NCCP 22456) and *Candida albicans* (NCCP 32703) biofilms. Biofilms were subjected to Nd:YAG (fluence 200 mJ/cm², pulse duration, 0.1 ms; spot size, 1.5 mm; frequency, 30 Hz) for cumulative time periods (0, 10, 20, and 60 s). Biofilm biomass, cell viability, and metabolic activity were assessed using crystal violet staining, colony counting, and tetrazolium salt (2,3-bis [2-methoxy-4-nitro-5-sulfophenyl]-2H-tetrazolium-5-carboxanilide; XTT) assay, respectively. Biofilm ultrastructure was assessed using scanning electron microscopy (SEM). Biofilm biomass significantly decreased after 10 s of laser irradiation in *T. rubrum* and *C. albicans* ($P < 0.01$), reaching maximum reductions of 77.1% and 58.1% after 60 s, respectively. Colony counts decreased by 66.0% for *T. rubrum* and 82.2% (0.75 log) for *C. albicans*; however, these reductions remained below 1 log (<90%), indicating limited viability inhibition. Metabolic activity significantly decreased after 10 s in both strains ($P < 0.01$), reaching a 45.1% reduction in *T. rubrum* and 72.5% in *C. albicans* after 60 s. SEM imaging showed damaged hyphae and extracellular matrix in *T. rubrum*, and shrunken and broken spores in *C. albicans*. Overall, Nd:YAG effectively inhibited biofilm biomass, metabolic activity, and structural integrity but had a limited effect on reducing biofilm viability.

**IMPORTANCE**   This study assessed the effects of neodymium-doped yttrium aluminum garnet (Nd:YAG) laser irradiation on *Trichophyton rubrum* and *Candida albicans* biofilms, including changes in biomass, viability, metabolic activity, and structure. Regarding *T. rubrum*, no previous *in vitro* study has comprehensively reported on these specific parameters following Nd:YAG exposure. Existing *in vitro* studies on *T. rubrum* have only assessed the "fungicidal" efficacy of the Nd:YAG laser; the most common method has been measuring the difference in "colony diameter" between irradiated and non-irradiated colonies, followed by scanning electron microscopy (SEM) imaging of damaged hyphae. For *C. albicans*, only one study has examined biofilm viability and ultrastructural changes after Nd:YAG irradiation. Therefore, in accordance with recent recommendations for biofilm research, we conducted a thorough assessment of the laser's antibiofilm activity. The antibiofilm efficacy of the Nd:YAG laser alone was suboptimal, supporting its potential use as an adjunctive therapy rather than a standalone treatment.

**KEYWORDS**   biofilm, Nd:YAG, *Trichophyton rubrum*, *Candida albicans*, onychomycosis

Address correspondence to Min Kyung Shin, haddal@hanmail.net.

The authors declare no conflict of interest.

Onychomycosis, a fungal infection of the nail unit, is treatment-resistant; up to 54% of cases experience treatment failure, and 10%–53% undergo recurrence (1, 2). The sessile biofilm microbial community embedded in the extracellular matrix (ECM) is regarded as a key virulence factor for antifungal drugs and host defense tolerance (3, 4). Compared to planktonic cells, fungal cells in biofilm are 1,000 times more tolerant to antifungal drugs (5, 6). Moreover, planktonic cells originating from the biofilm matrix exhibit greater cytotoxicity and immune tolerance (6–8). Accordingly, the development of new antibiofilm agents (9–13), verification of the antibiofilm efficacy of antifungal drugs (13–16), and preliminary disruption of biofilms before antifungal treatment have all been advocated to combat onychomycosis (9, 17).

Dermatophytes are the primary cause of onychomycosis, followed by non-dermatophyte molds and yeasts (18). While the pathophysiology of bacterial and yeast biofilms has been extensively studied owing to their potentially lethal infections in medical devices, such as catheters and implants, studies on dermatophyte biofilms are comparatively scarce (19, 20). *Trichophyton rubrum* is the predominant dermatophyte pathogen, accounting for 60%–70% of infections (21, 22). Although yeast nail infections are uncommon and usually limited to immunocompromised individuals, *Candida albicans* is responsible for approximately 60% of these infections (23, 24). Although both *T. rubrum* and *C. albicans* biofilms contribute to antifungal resistance, their pathophysiologies and morphologies are distinct from one another. *T. rubrum* is a filamentous fungus that forms dense hyphal networks along keratinized tissues (25, 26); the areas between the septate hyphae become interconnected with the ECM, forming a mesh-like biofilm architecture (19, 27, 28). *C. albicans* is a dimorphic yeast that transitions between yeast and hyphal forms (29); multiple cell types, including yeast cells, pseudohyphae, and true hyphae, form densely packed cell communities and are encapsulated by a polysaccharide-rich ECM, forming heterogeneous biofilm structures (29–31).

Although various fungal biofilm treatment strategies—enzymes chemically dissolving the outer ECM layer, dispersal agents preventing initial biofilm adherence and detaching cell communities, mechanical or light-based devices disrupting biofilm architecture, and nanoparticle-based drug delivery systems—have been explored (11, 19, 20, 32, 33), therapeutic options for onychomycosis remain limited. The dense keratinized structure of the nail plate restricts penetration of drugs, therapeutic agents, and mechanical energy into the underlying biofilm structures. To date, photodynamic therapy (PDT) has shown the most promising results (34–36). Treatment involves applying a photosensitizing agent to the nail, followed by light exposure (400–700 nm) to generate reactive oxygen species. Localized oxidative stress damages fungal biofilm components without harming the adjacent nail tissues. However, the clinical application of PDT in onychomycosis remains limited owing to the variability in treatment protocols and lack of large-scale clinical evidence (34, 35). Furthermore, treatment efficacy is reduced in chronic hyperkeratotic onychomycosis due to insufficient penetration of the photosensitizer (34, 36).

Neodymium-doped yttrium aluminum garnet (Nd:YAG) laser is one of the most common laser therapies used in onychomycosis treatment (37–40). Its main fungicidal mechanism is "selective photothermolysis" (38, 41–43). The 1,064 nm wavelength pulses emitted by the laser are absorbed by chromophores (xanthomegnin, chitin, and melanin) in the dermatophyte mycelia. Subsequently, light energy is converted into heat, resulting in fungal cell death. The longer wavelength of the Nd:YAG laser enables deeper penetration through the nail plate, which is advantageous for onychomycosis treatment. The US Food and Drug Administration approved its use for "aesthetic" improvement of discolored fungal nails, but not for "fungicidal" treatment of nail dermatophytosis due to the lack of clinical evidence.

The antibiofilm efficacy of Nd:YAG lasers remains unexplored. While the Nd:YAG laser has demonstrated inhibition of bacterial biofilms *in vitro* and is currently used for biofilm eradication in dentistry (44–49), its effects on fungal biofilm have not been comprehensively evaluated. Therefore, this *in vitro* study assessed the antibiofilm effect

of the Nd:YAG laser in *T. rubrum* and *C. albicans*, the most common dermatophyte and yeast strains causing onychomycosis (50, 51). We evaluated changes in biofilm biomass, viability, metabolic activity, and morphology after cumulative laser irradiation.

## MATERIALS AND METHODS

### Fungal strains and growth conditions

This study was performed using standard strains of *T. rubrum* (NCCP 22456) and *C. albicans* (NCCP 32703) obtained from the National Culture Collection for Pathogens (NCCP) of the Republic of Korea.

### *Trichophyton rubrum*

Fungal inocula were prepared as described previously (27, 28). Samples were cultured on Sabouraud dextrose agar (SDA) and incubated at 28℃ for up to 14 days. The agar plate was covered with 5 mL sterile saline, and the colony surface was scraped with a sterile swab. The swabs were suspended in phosphate-buffered saline (PBS) and allowed to settle at 28℃ for 5 min. The resulting solution was adjusted to an optical density (OD) of 0.65 at 570 nm, corresponding to a fungal concentration of $10^7$ colony-forming units (CFU)/mL.

### *Candida albicans*

Fungal inocula were prepared as previously described (52). Samples were cultured on SDA and incubated at 37℃ for 24 h. The colonies were scraped and transferred to yeast peptone dextrose (YPD) broth. The broth was sub-cultured at 37℃ for 18 h in a 130 rpm centrifuged shaker. The resulting fungal suspension was filtered through a cell strainer (VWR, USA; pore size 100 µm) to remove clumps and resuspended in 0.5 mL of YPD. The concentration was then adjusted to an OD of 0.65 at 570 nm for a final concentration of $10^7$ CFU/mL.

### Biofilm formation

### *Trichophyton rubrum*

Fungal biofilms were formed in 96-well flat-bottom microtiter plates as previously described (27, 53). A volume of 250 µL of the prepared *T. rubrum* suspension was pipetted into the microtiter plate. For the adhesion phase, the well plate was incubated for 3 h at 37℃ without agitation. After washing with 250 µL of PBS, 250 µL of RPMI 1064 medium was pipetted into each well. For the development phase, the plate was incubated for 72 h at 37℃ without agitation and then washed with 250 µL of PBS.

### *Candida albicans*

The biofilm formation assay was performed according to the method described by Pierce et al. with minor modifications (52). Previously prepared *C. albicans* suspension (250 µL) was pipetted into a 96-well flat-bottom microtiter plate. For the adhesion phase, the plate was incubated for 1.5 h at 30℃ in a shaker at 75 rpm. The well was then washed using 250 µL of PBS, and 250 µL of YPD was added. For the development phase, the plate was incubated for 48 h at 30℃ in a shaker at 75 rpm while the broth was changed every 24 h. Finally, the well was washed with 250 µL of PBS.

### 1,064 nm Nd:YAG laser irradiation

The biofilms formed in the microtiter plate were irradiated using a 1,064 nm Nd:YAG laser (PinPointe FootLaser, Pinpointe USA, Inc., Chico, CA, USA) at the following settings: fluence, 200 mJ/cm²; pulse duration, 0.1 ms; spot size, 1.5 mm; frequency, 30 Hz; and temperature, 40℃–60℃. The laser beam moved in a spiral pattern over the surface from

a distance of 5 mm. The duration of laser exposure was 0, 10, 20, or 60 s. The bottom of the well plate was monitored with an infrared thermometer to ensure the temperature did not exceed 42°C.

Preliminary experiments were conducted before the main experiment to determine the optimal laser exposure time and experimental conditions. These preliminary data were consistent, supporting the reliability and reproducibility of the data. Therefore, for each fungal strain, the main experiment was performed once in a single irradiation session. Each assessment included at least three technical replicates.

## Biofilm biomass quantification by CV staining

Biofilm biomass reduction after 1,064 nm Nd:YAG laser irradiation was evaluated using crystal violet (CV) staining as described previously (27, 28, 54). The laser-exposed well plates were stained using 200 µL of 0.5% CV solution for 10 min at room temperature. Excess stain was washed with distilled water, and the remaining CV was released using 33% acetic acid. The OD of each well was measured at 600 nm using a microplate reader (Spark 10M, Tecan Group Ltd., Männedorf, Switzerland).

## Biofilm viability evaluation by colony counting

Colony counting was used to evaluate the viability of fungal strains after exposure to the Nd:YAG laser. Due to the distinct growth patterns of the two organisms, different plating methods were employed. While the conventional spread-plate method was used for *C. albicans*, a modified version of the drop-plate method was used for *T. rubrum* (55, 56). When we initially applied the spread-plate method to *T. rubrum*, colonies tended to merge, making CFU counting inaccurate, owing to their radial mycelial growth pattern. Therefore, to prevent colony merging during incubation, we placed small droplets of *T. rubrum* suspension onto the growth medium, spaced at least 0.5 cm apart. After incubation, the colony-forming droplets were counted. Compared to the original drop-plate method, where colonies forming within a single droplet were counted to calculate CFU/mL (55), this study only assessed whether at least one fungal colony had formed within each droplet. This modification was necessary for the same reason we adapted the drop-plate method initially: *T. rubrum* colonies within a single droplet merged together, making individual colony counts impossible.

### *Trichophyton rubrum*

For *T. rubrum*, 200 µL of PBS was added to each previously laser-exposed well, and the well bottom was scraped with a sterile spatula. Fungal suspensions from three technical replicates (wells) were mixed in a single Falcon tube without additional dilution. Biofilms were disrupted using an ultrasonic homogenizer (30 s, 50 W). The resulting suspension was placed on SDA as 15 individual droplets (each 15 µL), spaced at least 0.5 cm apart, and incubated for up to 4 days in a 28°C humidified incubator. The number of colony-forming droplets was then determined.

### *Candida albicans*

The bottom of each irradiated well was scraped with a sterile spatula, and the material was transferred to a Falcon tube containing 10 mL of PBS (dilution factor: $10^{-1}$). Biofilms were disrupted using an ultrasonic homogenizer (30 s, 50 W). Subsequently, 100 µL of the suspension was spread on SDA and incubated for 24 h at 37°C. The resulting colonies were counted.

## Biofilm metabolic activity assessment by the XTT assay

The 2,3-bis [2-methoxy-4-nitro-5-sulfophenyl]-2H-tetrazolium-5-carboxanilide (XTT) assay was used to evaluate biofilm metabolic activity as described previously (27, 52). The laser-exposed plates were washed with PBS. Then, 70 µL of the XTT solution (0.2

mg/mL) from the XTT kit (Thermo Fisher Scientific, USA) was added to each well and incubated at 38°C for 2 h in the dark. Absorbance at 450 nm was measured using a microplate reader (E-max, Molecular Devices, Sunnyvale, CA, USA).

## Biofilm and fungal ultrastructure imaging by SEM

For scanning electron microscopy (SEM) examination, fungal biofilms were formed on 24-well microtiter plates containing a 12 mm coverslip (Flux) as described previously (27, 57). After laser exposure, the coverslips were transferred to a desiccator containing silica gel and air-dried at room temperature for 3 days. The samples were dehydrated with a critical-point dryer and platinum-coated via ion sputtering (Hitachi E-1045, Hitachi, Tokyo, Japan). Images were acquired using a field-emission SEM (S-4800, Hitachi, Tokyo, Japan). The image acquisition parameters were those that produced the sharpest images, as assessed visually. At least ten images were analyzed for each laser exposure group.

## *T. rubrum* biofilm autofluorescence assessment by CLSM

Confocal laser scanning microscopy (CLSM) was used to assess the autofluorescence of *T. rubrum* biofilms after Nd:YAG irradiation. Endogenous fluorophores—such as nicotinamide adenine dinucleotide, flavins, and melanin—are present within the hyphae and microconidia of filamentous fungi (58–60), enabling label-free visualization of *T. rubrum* structures. Consequently, changes in autofluorescence intensity can reflect laser-induced alterations in biofilm structure and cellular integrity (59, 61). An identical *T. rubrum* suspension used for colony counting was prepared for CLSM imaging. The fungal suspension was seeded on a cell culture slide (SPL Life Sciences, Republic of Korea) and incubated for up to 7 days in a 28°C humidified incubator. Autofluorescence imaging was performed using a confocal laser scanning microscope (LSM 700, Carl Zeiss, Wetzlar, Germany). The microscope was equipped with an excitation wavelength of 405 nm, and the emitted fluorescence was collected within 461–465 nm, primarily detecting intracellular NAD(P)H, an endogenous fluorescent cofactor involved in cellular metabolism (58). Corresponding bright-field images were acquired simultaneously to visualize fungal structures. At least ten images were analyzed for each laser exposure group. CLSM imaging was not performed for *C. albicans* because its intrinsic autofluorescence was insufficient for reliable visualization of endogenous fluorophores under the selected imaging conditions.

## Statistical analysis

The means of the technical replicates were used for statistical evaluation. One-way analysis of variance, followed by post hoc Bonferroni and Kruskal-Wallis tests, was used to evaluate biofilm inhibition after various laser exposure times. Correlation analysis was performed using Spearman's rho test. Statistical analyses were performed using SPSS software (version 22.0, SPSS Inc., Chicago, IL, USA). A $P$ value < 0.05 was considered statistically significant.

## RESULTS

### Effect of 1,064 nm Nd:YAG on *T. rubrum* and *C. albicans* biofilm biomass

Biofilm inhibition following 1,064 nm Nd:YAG laser exposure was quantified using CV staining. The resulting OD values after different laser exposure times (10, 20, and 60 s) were compared to those of the control (Fig. 1). Both *T. rubrum* and *C. albicans* showed a significant decrease ($P < 0.01$) in biofilm biomass after 10 s of laser exposure. *T. rubrum* displayed a strongly significant ($P < 0.001$) reduction in biomass for irradiation exceeding 10 s, while *C. albicans* did so for 20 s. *T. rubrum* achieved over 50% reduction in biofilm biomass after 10 s and *C. albicans* after 20 s of laser exposure. The final percentage reduction in biofilm biomass after 60 s of laser exposure was 77.1% for *T. rubrum* and

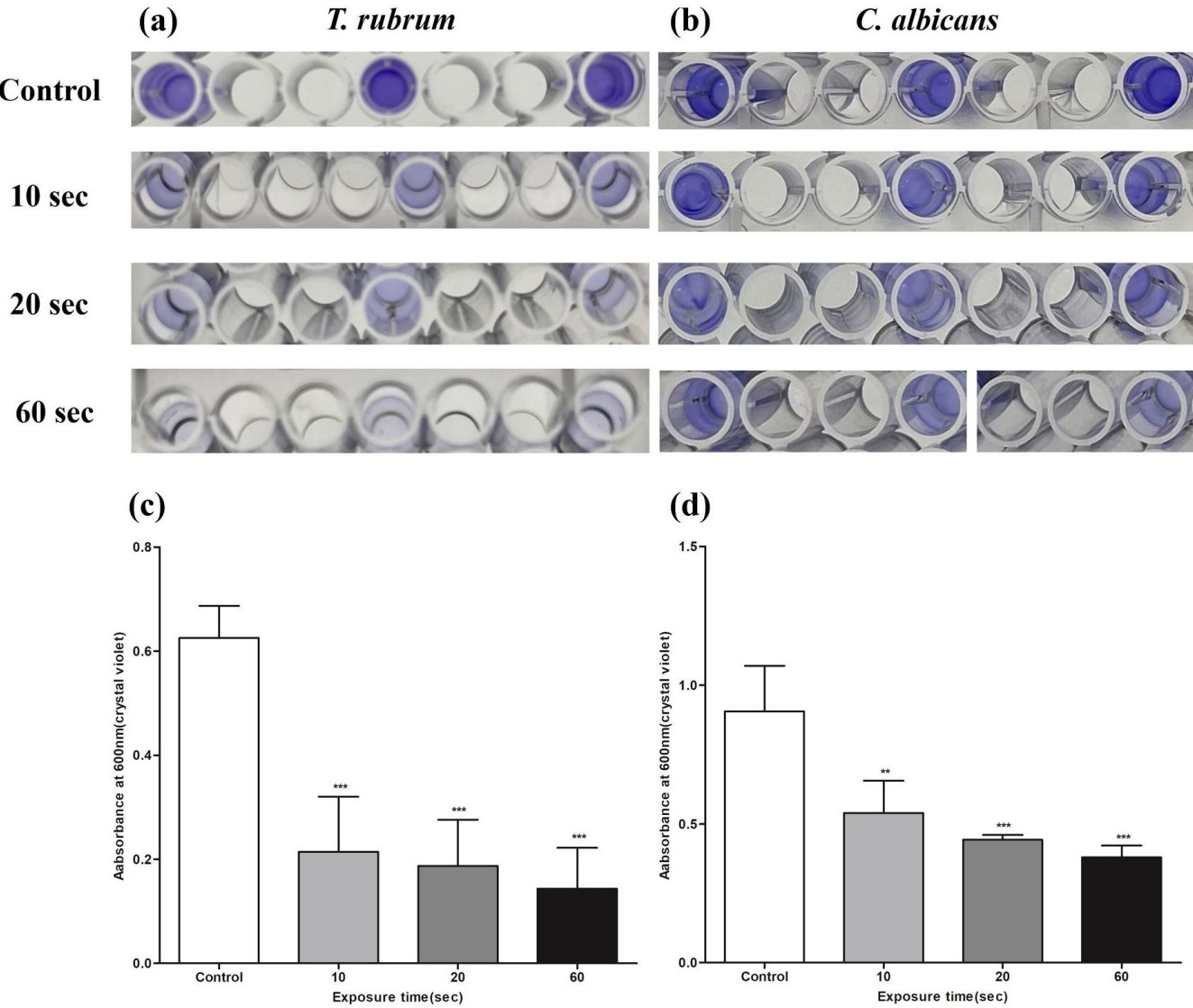

**FIG 1** Biofilm biomass quantification of *T. rubrum* and *C. albicans* upon 1,064 nm Nd:YAG laser exposure using crystal violet staining (OD 600 nm). Visualization of (a) *T. rubrum* and (b) *C. albicans* biofilms on 96-well microtiter plates according to the laser exposure time. The OD readings of both (c) *T. rubrum* and (d) *C. albicans* show significantly decreased biofilm biomass after 10 s of laser exposure compared to the control. Data are shown as the mean ± SD. **$P < 0.01$; ***$P < 0.001$.

58.1% for *C. albicans*. Additionally, the susceptibility to laser irradiation increased with exposure time. Both fungal strains showed a significant negative correlation between laser exposure time and biofilm biomass: $r = −0.78$ ($P < 0.001$) for *T. rubrum*, $r = −0.86$ ($P < 0.001$) for *C. albicans*.

## Effect of 1,064 nm Nd:YAG on *T. rubrum* and *C. albicans* biofilm viability

Biofilm viability after Nd:YAG laser exposure was assessed using a modified drop-plate colony counting method for *T. rubrum* and a conventional spread-plate colony counting method for *C. albicans*. Both fungal strains showed significantly decreased ($P < 0.01$) colony counts after 20 s of laser exposure compared to the control (Fig. 2); the significance was stronger in the 60 s exposed group ($P < 0.001$). After 60 s of laser exposure, *T. rubrum* achieved a 66.0% reduction in colony-forming droplet count, and *C. albicans* 82.2% (0.75 log) reduction in CFU count. The percentage of colony count reduction was less than 90% (1 log CFU) for both strains, indicating a limited reduction in biofilm viability. Both strains displayed a negative correlation between the colony count and

laser exposure time: *r* = −0.95 (*P* < 0.001) for *T. rubrum* and *r* = −0.92 (*P* < 0.001) for *C. albicans*.

## Effect of 1,064 nm Nd:YAG on *T. rubrum* and *C. albicans* biofilm metabolic activity

The inhibition of biofilm metabolic activity following 1,064 nm Nd:YAG exposure was measured using the XTT assay (Fig. 3). Both *T. rubrum* and *C. albicans* showed a significant decrease (*P* < 0.01) in biofilm metabolic activity after 10 s of laser exposure. *T. rubrum* displayed a strong significance (*P* < 0.001) for irradiation exceeding 20 s, while *C. albicans* did so for 10 s. *T. rubrum* failed to achieve 50% reduction in biofilm metabolic activity, while *C. albicans* succeeded after 20 s of irradiation. The ultimate percentage reduction after 60 s of laser exposure was 45.1% for *T. rubrum* and 72.5% for *C. albicans*. Both fungal strains showed a significant negative correlation between laser exposure time and biofilm metabolic activity: r = −0.78 (*P* < 0.001) for *T. rubrum* and r = −0.88 (*P* < 0.001) for *C. albicans*.

## Effect of 1,064 nm Nd:YAG on *T. rubrum* biofilm ultrastructure

SEM was used to examine the morphological modifications of fungal structures after 1,064 nm Nd:YAG exposure. Figure 4 shows progressive malformations in the hyphae, cell walls, and biofilm structure of *T. rubrum* with increasing laser exposure time. The control group showed uniform tubular hyphae with intact cell walls and blanket-like biofilms interconnecting the adjacent hyphae (Fig. 4a and b). The 10 s exposed group displayed broken and wrinkled cell walls, resulting in a partially ruptured ECM (Fig. 4c and d). The 20 s group showed scattered and fragmented hyphae (Fig. 4e and f), and the 60 s group showed clumped hyphae with intracellular material leakage (Fig. 4g and h). Similarly, CLSM imaging showed a subsequent decrease in autofluorescence (Fig. 5a, c, and e) and hyphal density (Fig. 5b, d, and f) with increasing laser exposure time, indicating inhibition of cellular metabolism and hyphal growth within the biofilm.

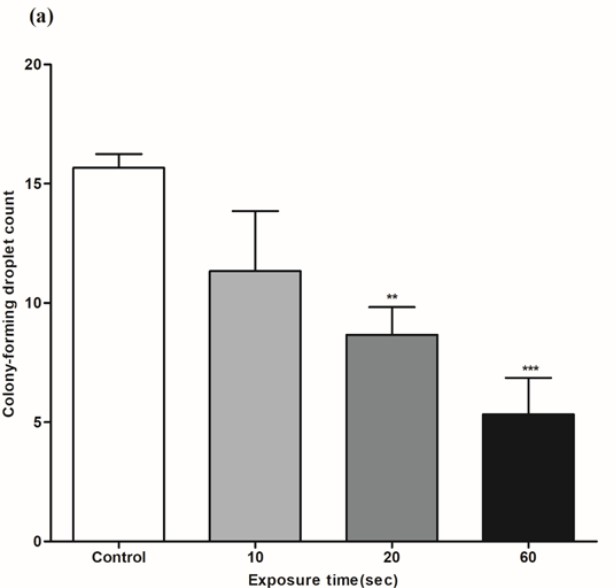
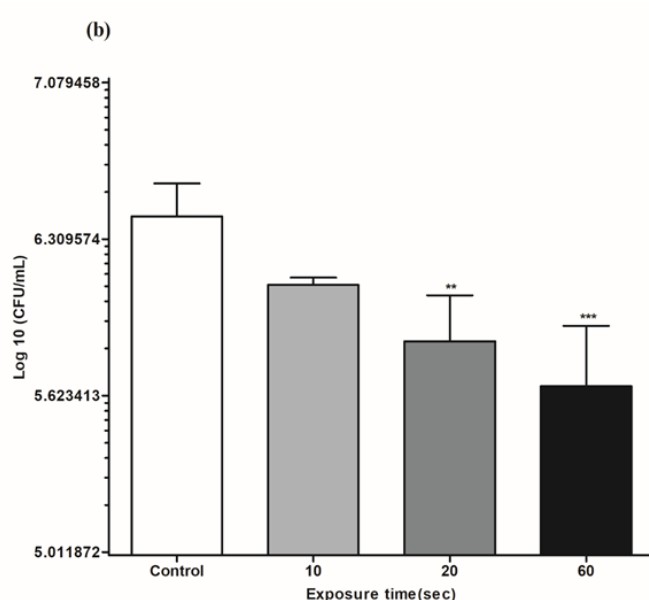

**FIG 2**  Biofilm viability evaluation of *T. rubrum* and *C. albicans* upon 1,064 nm Nd:YAG laser exposure using colony counts. Both (a) *T. rubrum* and (b) *C. albicans* show significantly decreased colony counts after 20 s of laser exposure compared with the control. However, reductions remained <1 log CFU (<90%), indicating limited inhibition of biofilm viability. Data are shown as the mean ± SD. **P < 0.01; ***P < 0.001.

## Effect of 1,064 nm Nd:YAG on *C. albicans* biofilm ultrastructure

SEM images of *C. albicans* showed the cumulative deformation of yeast cells and their surrounding biofilms with increasing laser exposure time (Fig. 6). The control group showed well-rounded yeast cells with intact cell membranes, and the ECM encapsulating the cells was well-preserved (Fig. 6a and b). However, after 10 s of laser irradiation, the cell surfaces began to wrinkle (Fig. 6c), and the magnified images showed ECM breakage (Fig. 6d). After 20 s of exposure, the cell surfaces were further fragmented, and the intracellular material leaked (Fig. 6e and f). Finally, perforated and swollen cells aggregated into amorphous clusters in the 60 s group (Fig. 6g and h).

## DISCUSSION

This study aimed to assess the antibiofilm activity of Nd:YAG laser against *T. rubrum* and *C. albicans* biofilms. We evaluated biofilm biomass reduction by CV staining, biofilm viability loss by colony counting, metabolic activity decline by XTT assay, and structural disruption by SEM. Both *T. rubrum* and *C. albicans* showed significant reductions in biofilm biomass, metabolic activity, and structural integrity, but only limited inhibition of biofilm viability. Antibiofilm efficacy positively correlated with laser irradiation time, demonstrating energy dependence.

To the best of our knowledge, this is the first study to comprehensively assess the antibiofilm effects of Nd:YAG lasers in onychomycosis. Regarding *T. rubrum*, no previous *in vitro* study has reported the biofilm biomass, viability, metabolic activity, and structural modifications caused by Nd:YAG exposure. The existing *in vitro* studies on *T. rubrum* have only assessed the "fungicidal" efficacy of the Nd:YAG laser; the difference in "colony diameter" between laser-irradiated and non-irradiated *T. rubrum* colonies has been the most commonly evaluated method (62–68), followed by SEM imaging of damaged hyphae (62, 67, 69). For *C. albicans*, only one study examined biofilm viability and ultrastructural changes after Nd:YAG irradiation (70); the biofilm CFU count decreased by 52%; however, the biofilm architecture showed no meaningful modifications in SEM imaging. Furthermore, this study properly assessed the antibiofilm effects utilizing both traditional and up-to-date methodologies. Previous dermatophyte antibiofilm studies have generally incorporated two to three biofilm assessment methods (19, 71), with the

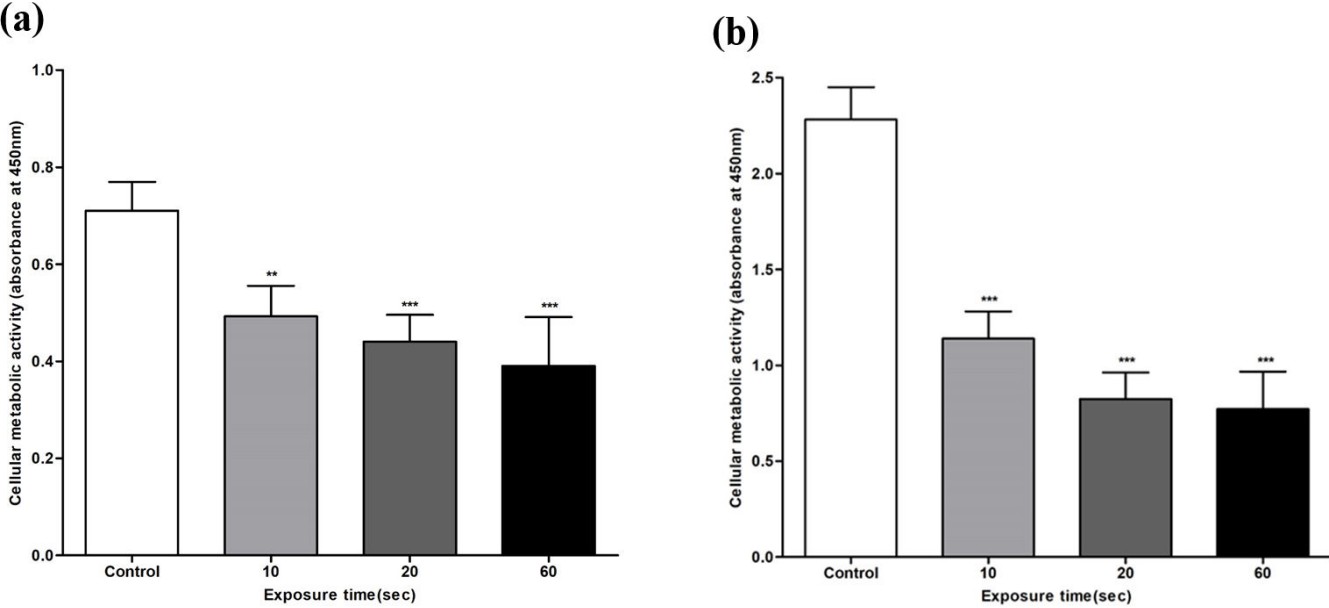

**FIG 3** Biofilm metabolic activity assessment of *T. rubrum* and *C. albicans* upon 1,064 nm Nd:YAG laser exposure using XTT assay (OD 450 nm). OD readings of (a) *T. rubrum* and (b) *C. albicans* show significantly decreased biofilm metabolic activity after 10 s of laser exposure compared to the control. A strong significance is observed for irradiation exceeding 20 s for *T. rubrum* and 10 s for *C. albicans*. Data are shown as the mean ± SD. **$P < 0.01$; ***$P < 0.001$.

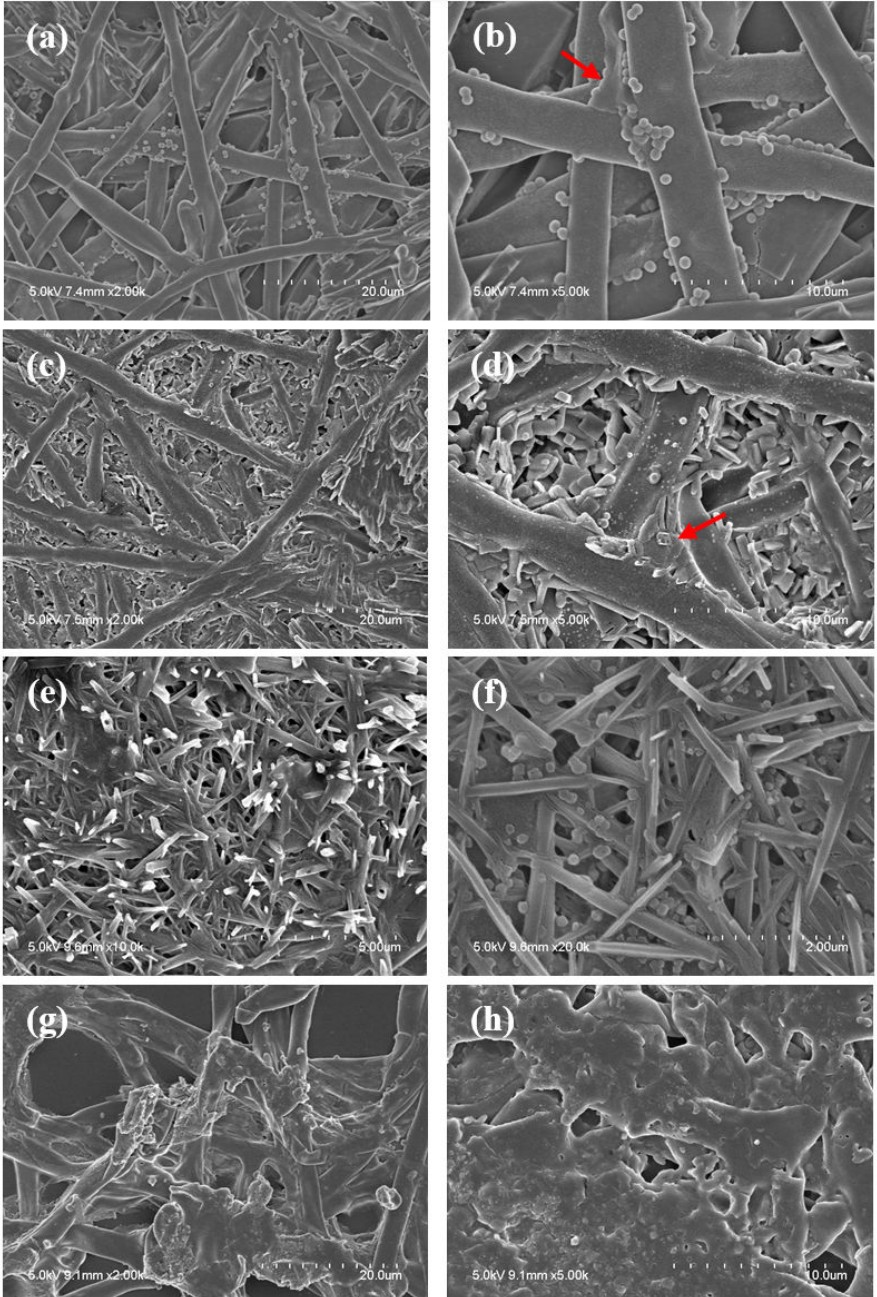

**FIG 4** Scanning electron microscopic images of *T. rubrum* upon 1,064 nm Nd:YAG laser exposure. (a) The control group exhibits uniform tubular hyphae with intact cell walls. (b) Blanket-like biofilms (red arrows) interconnect the adjacent hyphae. (c and d) The 10 s exposed group displays wrinkled cell walls and ruptured ECM (red arrow). (e and f) Hyphae are further scattered and fragmented in the 20 s group. (g and h) Finally, the 60 s exposed group displays clumped hyphae with intracellular material leakage, transforming into flat structures.

XTT assay, SEM imaging, and CV assay being the most common. This study used four evaluation methods: CV assay, colony count, XTT assay, and SEM imaging. Therefore, based on recent biofilm study recommendations (13, 72, 73), we comprehensively assessed the antibiofilm ability of the Nd:YAG laser.

Despite the differences in biofilm morphology, composition, and metabolism between dermatophytes and yeasts, the antibiofilm efficacy of the Nd:YAG laser was similar between *T. rubrum* and *C. albicans*. Filamentous fungi undergo hyphal growth and

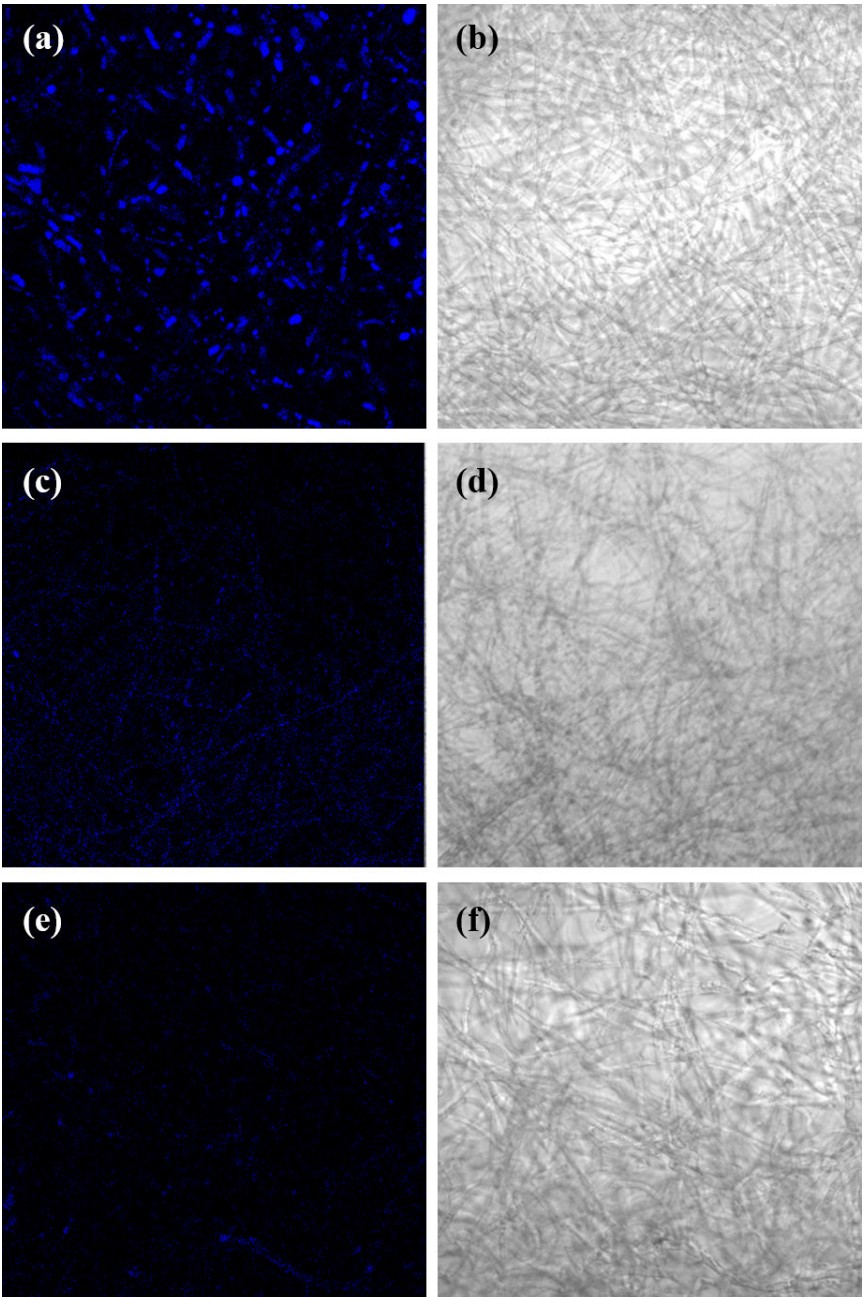

**FIG 5** Confocal laser scanning microscopy images of *T. rubrum* upon 1,064 nm Nd:YAG laser exposure. The control group exhibits (a) dense fungal autofluorescence and (b) hyphal formations. In accordance with the laser exposure time, fungal autofluorescence and hyphal density subsequently decrease in the (c and d) 20 s and (e and f) 60 s laser-exposed groups. Images were captured at a magnification of ×400.

lack the asexual phase progression (budding and binary) present in yeast and bacteria (25, 26). Therefore, their biofilms are mainly composed of interconnected hyphal bundles rather than the cell microcolonies observed in yeasts (74–76). SEM imaging in this study clearly demonstrated this difference: *T. rubrum* showed a dense hyphal network with blanket-like ECM, and *C. albicans* round yeast cells with ECM encapsulation (Fig. 4 and 5). Regardless of these structural differences, the Nd:YAG laser sufficiently disrupted the fungal cell membranes and ECM of both species in this study. Secondly, important ECM components are dissimilar between filamentous fungi and yeasts. Although galactosami-nogalactan (GAG) is an integral polysaccharide in filamentous fungi biofilm (75, 77), the

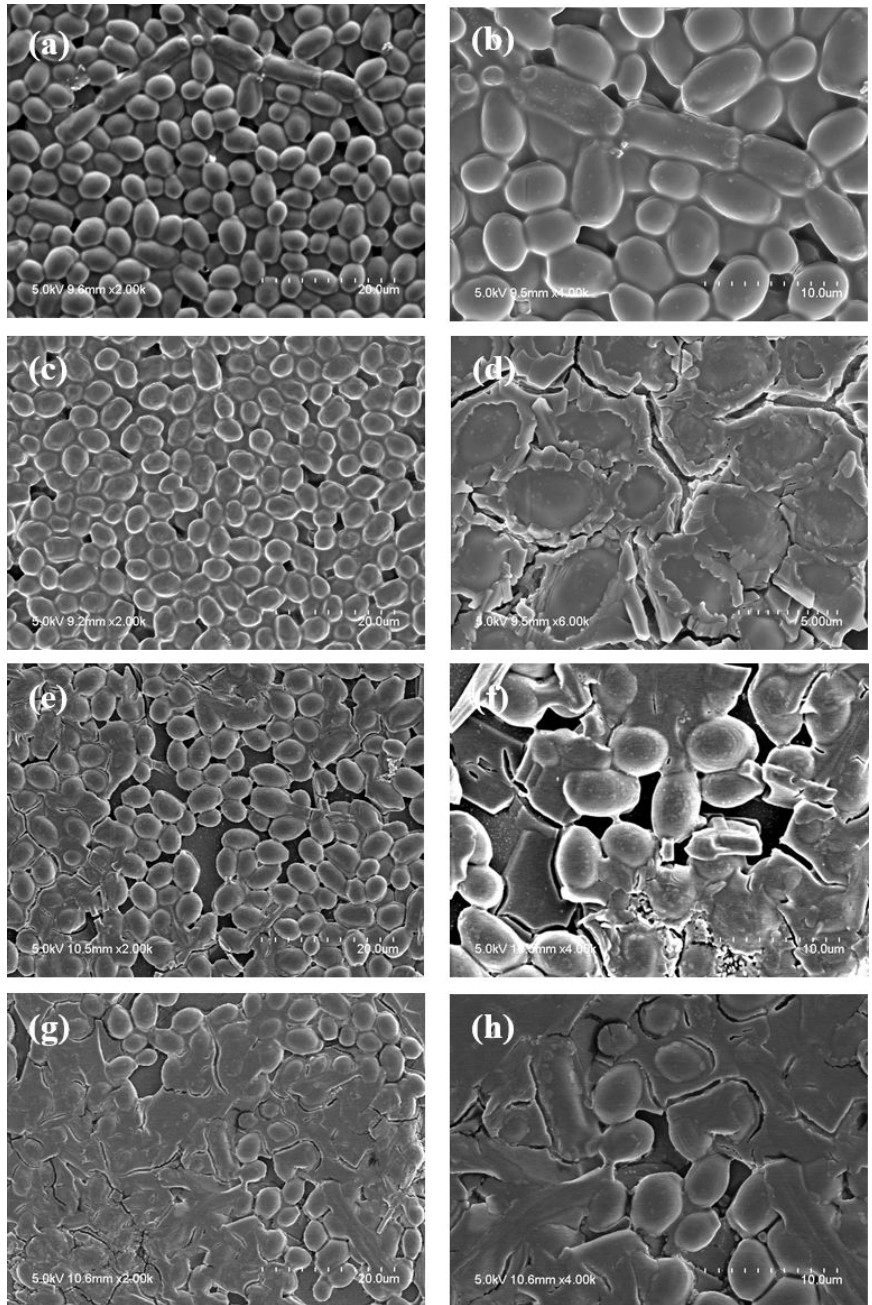

**FIG 6**  Scanning electron microscopy of C. *albicans* upon 1,064 nm Nd:YAG laser exposure. (a and b) The control group shows well-rounded yeast cells, intact cell membranes, and a smooth ECM encapsulation. (c) After 10 s of laser irradiation, the cell surface wrinkles, resulting in shrunken cells. (d) Magnified images reveal broken ECM. (e and f) The 20 s exposure group shows broken cell walls and intracellular material leakage. (g and h) Finally, the perforated cells are swollen and aggregated into clusters in the 60 s-exposed group.

GAG synthesis gene cluster is absent in *C. albicans* (75). Similarly, sterols and sphingolipids are crucial ECM components responsible for antifungal resistance in *C. albicans*, whereas their synthesis is minimal in filamentous fungi (74, 78). Additionally, only a few transcriptional regulators involved in filamentous fungal biofilms are present in *C. albicans* (75). Such compositional and metabolic differences may explain the biofilm biomass and metabolic activity observed in this study. Although the biofilm biomass significantly decreased after 10 s of laser exposure in both fungal strains, *T. rubrum*

showed greater biomass inhibition within the same irradiation time (Fig. 1). Metabolic activity declined significantly after 10 s of laser exposure in both fungal strains; however, *C. albicans* exhibited greater metabolic inhibition (Fig. 3). While *T. rubrum* failed to achieve 50% reduction in metabolic activity at all exposure times, *C. albicans* succeeded within 10 s of laser exposure. Further studies on "omics" analyses—genomics, transcriptomics, proteomics, and metabolomics—after Nd:YAG irradiation are necessary to explain such discrepancies. Nevertheless, the capacity of the Nd:YAG laser to inhibit both dermatophyte and yeast biofilms highlights its advantages as a broad-spectrum antibiofilm modality.

The antibiofilm effect of the Nd:YAG laser was energy-dependent for both *T. rubrum* and *C. albicans*; biofilm biomass, cell viability, metabolic activity, and structural integrity significantly decreased with laser exposure time ($P < 0.01$). Although previous studies have reported a positive correlation between "clinical efficacy" and "mycological growth inhibition" with increasing laser energy in *T. rubrum* and *C. albicans* (62, 79–83), the changes in antibiofilm activity remain unexplored. Hence, this study showed that sufficient Nd:YAG irradiation enhanced the antibiofilm efficacy in fungal biofilms.

The selection of a nanosecond or picosecond pulse duration for the Nd:YAG laser was not necessary to achieve antibiofilm efficacy. Nd:YAG lasers inhibit fungal growth via three mechanisms: photothermal, photomechanical, and photochemical. Among these, "selective photothermolysis" is regarded as the key mechanism underlying its fungicidal effect (18, 38, 41–43). Chromophores, such as xanthomegnin, chitin, and melanin in dermatophyte mycelia, absorb laser energy (84–87). Light energy is further converted into heat, and elevated temperatures kill the fungal cells. To limit thermal damage to the adjacent normal tissues, the pulse duration must be shorter than the thermal relaxation time of the fungus (84, 88, 89). The thermal relaxation time for dermatophyte structures is as follows: 0.004–0.1 ms for hyphae, 16 µs to 2.5 ms for macroconidia, and 0.004–0.016 ms for microconidia (84). Therefore, selective photothermolysis requires nanosecond or picosecond-range pulse durations. Previous Nd:YAG laser studies demonstrating antibiofilm efficacy in bacterial microbes mostly used nanosecond-range Q-switched Nd:YAG laser (44, 45, 48, 49). By contrast, this study used a PinPointe FootLaser with a 0.1 ms pulse duration. Therefore, this study shows that a wide range of conventional Nd:YAG lasers with millisecond-, nanosecond-, and picosecond-range pulse durations can be used as antibiofilm modalities for onychomycosis.

Reaching the fungicidal temperature was not required to achieve the antibiofilm effect of the Nd:YAG laser. Because fungi are heat sensitive and Nd:YAG laser increases the target temperature, thermal damage has been acknowledged as the main fungicidal mechanism of laser treatment (62, 87). Temperatures exceeding 50°C–60°C have been reported to inhibit the growth of *T. rubrum* and *C. albicans* (66, 88, 90). Accordingly, *in vitro* studies have claimed that reaching the fungicidal temperature using an Nd:YAG laser is the key factor for sufficient *T. rubrum* growth inhibition (43, 66, 81). Nevertheless, this study demonstrated significant antibiofilm activity while maintaining a temperature below 42°C during irradiation.

Consequently, the Nd:YAG laser can inhibit fungal biofilms without injuring the adjacent normal tissues. The fungicidal temperatures of *T. rubrum* and *C. albicans* inevitably damage surrounding tissues. *In vivo* studies have shown that temperatures exceeding 43°C damage fibroblasts and 48°C induce burn injuries (91, 92). However, limiting the thermal damage to fungal cells is nearly impossible for two reasons. First, the 1,064 nm light energy is not only absorbed by chromophores in fungal mycelia, but also water in nearby tissues, resulting in "nonspecific bulk heating" of the irradiated area (84). Second, most Nd:YAG lasers used in onychomycosis treatment possess pulse durations longer than the thermal relaxation time of the fungus (40, 84), increasing the possibility of photothermal damage to nearby tissues. Therefore, because the Nd:YAG laser can inhibit biofilm formation without raising the temperature above 42°C, patients can safely receive antibiofilm therapy.

Non-fungal microbes have also exhibited antibiofilm activity following Nd:YAG irradiation in previous studies. Biofilms in dental peri-implantitis (15 bacterial strains) (44, 45), endodontic infections (*Enterococcus faecalis*) (46, 47), and heritage stone monument contamination (algae and cyanobacteria) (48, 49) have been successfully inhibited by Nd:YAG irradiation. In detail, the multispecies biofilm on the titanium surface was completely eliminated, showing a "biofilm-free" surface in the CV assay and "zero" total bacterial count (0.000 CFU/mL) in the CFU count (44, 45). The Nd:YAG irradiated *E. faecalis* biofilms showed a significantly decreased CFU count in the *in vitro* assay and a non-significant decline in the *ex vivo* assay (46, 47). The subaerial biofilm, composed of bacterial and algal species, on granite stone was significantly decreased in the stereomicroscopic analysis (48, 49). The authors commonly praised the ability of Nd:YAG lasers to decrease the biofilm bioburden while minimizing surface damage (44, 48). These results highlight how Nd:YAG lasers can disrupt the biofilm architecture in onychomycosis without harming the adjacent nail apparatus.

Compared with PDT, the Nd:YAG laser displays inferior antibiofilm efficacy. Photodynamic therapy is one of the most well-established treatments for fungal biofilms (11, 34, 93, 94). Numerous *in vitro* and clinical studies have demonstrated its ability to inhibit biofilms of fungal pathogens (11, 12, 34, 53, 93). In case of *T. rubrum*, PDT caused 2.0–4.6 log CFU (53, 93), 70%–90% XTT assay (12, 93), and fourfold CV assay reduction (93). Similarly, SEM images display cell wall collapse, hyphal shrinkage, and ECM loss (12, 93). Compared to the *T. rubrum* results of this study—66% colony count, 45% XTT assay, and fourfold (77%) CV assay reduction—PDT showed superior antibiofilm activity compared to the Nd:YAG laser. For *C. albicans*, previous studies showed PDT inducing 1.0–1.4 log CFU and 45% CFU reduction (95–97), 57%–70% XTT decline (98–100), and microscopic cell wall damage (101). Comparatively, *C. albicans* results of this study—0.75 log CFU, 72% XTT assay, and twofold (58%) CV assay reduction—showed inferior biofilm viability reduction. Hence, the results of this study indicate that the Nd:YAG laser alone is insufficient to eradicate fungal biofilms.

However, the Nd:YAG laser can be used as an adjuvant antibiofilm modality for onychomycosis. Published studies have consistently reported significantly higher mycological and clinical cure rates with combined treatment—Nd:YAG laser with traditional antifungal drugs (systemic or topical)—than with laser or drug monotherapy (40, 88, 102–107). These results correlate with *in vitro* studies showing a significant reduction in the MICs of antifungal drugs (108). In contrast, systematic reviews of Nd:YAG laser monotherapy have reported inconsistent results, showing both promising and poor onychomycosis cure rates (39–41). When the review was limited to randomized controlled trials, the results showed limited efficacy (42). Similarly, *in vitro* studies assessing colony growth inhibition and hyphal structure damage by Nd:YAG laser monotherapy have reported conflicting results (62–69, 81, 109). Therefore, combining *in vitro* results of this study—presenting significant biofilm biomass, metabolic activity, and structural integrity inhibition—and previous studies reporting superior clinical and *in vitro* onychomycosis cure rates when used as an adjuvant therapy, the Nd:YAG laser can be used in conjunction with other fungicidal treatments to improve biofilm-induced treatment resistance.

The present study has several limitations. First, *ex vivo* and *in vivo* studies more accurately reflect the environmental conditions of the nail unit. *Ex vivo* studies show *T. rubrum* forming a more "robust" biofilm and fungal microbes obtaining higher antifungal resistance compared to *in vitro* assays (28, 110, 111). *In vivo* studies on *C. albicans* have shown that the majority of biofilm matrix proteins originate from the host, resulting in biofilm content variation in different hosts (112). Additionally, *in vivo* studies are necessary to develop appropriate treatment protocols (13, 113). Although this study demonstrated sufficient biofilm inhibition after 20 s of laser exposure, *in vivo* studies will require a longer treatment time; the thick nail apparatus acts as a physical barrier, and a larger infection area restricts laser energy accumulation. Therefore, further *ex vivo* and *in vivo* studies are required to understand biofilm-host interactions and develop treatment

standards. Second, lasers with nanosecond- or picosecond-range pulse durations, such as the Q-switched Nd:YAG laser, may yield superior antibiofilm results. A pulse duration shorter than the thermal relaxation time of the fungal structure enables efficient heat accumulation within the cell wall. Although this study demonstrated biofilm inhibition with a millisecond-pulsed laser, further comparisons with nanosecond-range Nd:YAG lasers are required.

In conclusion, the results of this study highlight the potential of Nd:YAG laser as an adjuvant antibiofilm modality for onychomycosis. Despite the limited reduction in biofilm viability, the biofilm biomass and metabolic activity were significantly inhibited in *T. rubrum* and *C. albicans*. Additionally, the biofilm ultrastructure was disrupted in both fungal strains. Although the antibiofilm efficacy of the Nd:YAG laser was not superior to that of PDT, its ability to penetrate hyperkeratotic nail tissue is advantageous for onychomycosis biofilm treatment. Further *ex vivo* and *in vivo* studies are necessary to investigate whether antibiofilm efficacy is maintained under realistic environmental conditions and to develop standard antibiofilm therapy protocols.

## AUTHOR AFFILIATIONS

[1]Department of Dermatology, School of Medicine, Kyung Hee University, Seoul, Republic of Korea
[2]Medical Science Research Institute, Kyung Hee University Medical Center, Seoul, Republic of Korea

## AUTHOR ORCIDs

Ji-In Seo http://orcid.org/0000-0002-7431-0816
Jin-Woo Lee http://orcid.org/0000-0003-0390-7954
Min Kyung Shin http://orcid.org/0000-0001-9834-7931

## AUTHOR CONTRIBUTIONS

Ji-In Seo, Conceptualization, Data curation, Formal analysis, Investigation, Visualization, Writing – original draft | Jin-Woo Lee, Data curation, Formal analysis, Investigation, Visualization | Min Kyung Shin, Conceptualization, Methodology, Resources, Supervision, Validation, Writing – review and editing

## ETHICS APPROVAL

The authors confirm that the ethical policies of the journal, as described in the author guidelines, have been adhered to. No ethical approval was required as the study involved only microorganisms.

## ADDITIONAL FILES

The following material is available online.

### Open Peer Review

**PEER REVIEW HISTORY (review-history.pdf).** An accounting of the reviewer comments and feedback.

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
