## [Reviewer comments · Microbiology Spectrum]

Microbiology Spectrum

Antibiofilm Efficacy of 1064-nm Neodymium-Doped Yttrium Aluminum Garnet (Nd:YAG) Laser on *Trichophyton rubrum* and *Candida albicans*: *in Vitro* Study

JIIN SEO, Jin-Woo Lee, and Min Shin

Corresponding Author(s): Min Shin, Kyung Hee University Hospital

Review Timeline:

Submission Date:	January 9, 2026
Editorial Decision:	January 30, 2026
Revision Received:	March 21, 2026
Accepted:	April 3, 2026

Editor: Luis Martinez

Reviewer(s): The reviewers have opted to remain anonymous.

Transaction Report:

DOI: <https://doi.org/10.1128/spectrum.00091-26>

Re: Spectrum00091-26 (Antibiofilm efficacy of 1064-nm Neodymium-Doped Yttrium Aluminum Garnet (Nd:YAG) laser on *Trichophyton rubrum* and *Candida albicans*: in vitro study)

Dear Dr. Min Kyung Shin:

Thank you for the privilege of reviewing your work. Below you will find my comments, instructions from the Spectrum editorial office, and the reviewer comments.

Your manuscript requires an extensive revision. You should emphasize in improving the clarity of the text and increase the technical replicates of the experiments. Also, present each replicate in each bar graph using individual symbols. Rigor is a concern and a fundamental importance for the journal. Please follow carefully and respond to each of the reviewer's recommendations since acceptance would depend on improving the manuscript significantly. It is likely that I will send it back to the reviewers for their approval of the changes.

Revision Guidelines

Sincerely,
Luis Martinez
Editor
Microbiology Spectrum

Reviewer #1 (Comments for the Author):

Overall, the English language requires careful revision, as the manuscript does not consistently employ appropriate scientific and academic terminology.

In general, the study presents methodologies that are applicable and consistent with the proposed experimental design. In particular, the methods employed to assess the outcomes are robust, and the experimental procedures are adequately described. However, the magnitude of the reductions reported throughout the manuscript appears to be overemphasized, especially in the Discussion section.

Although biofilms are well known to be complex and resilient structures that increase microbial resistance by limiting the effectiveness of therapeutic approaches (Li Y, Sun G, Xie J, Xiao S, Lin C. Antimicrobial photodynamic therapy against oral biofilm: influencing factors, mechanisms, and combined actions with other strategies. *Front Microbiol.* 2023 Jun 9;14:1192955. doi: 10.3389/fmicb.2023.1192955.), previous studies have reported more pronounced reductions in microbial viability when phototherapies are applied, particularly classical photodynamic therapy employing a photosensitizing agent. (Chen B, Sun Y, Zhang J, Chen R, Zhong X, Wu X, Zheng L, Zhao J. In vitro Evaluation of Photodynamic Effects Against Biofilms of Dermatophytes Involved in Onychomycosis. *Front Microbiol.* 2019 Jun 7;10:1228. doi: 10.3389/fmicb.2019.01228.). In this context, a more comprehensive discussion comparing the observed results with those obtained from conventional photodynamic therapy approaches-employing a photosensitizer in combination with a LED light source (including near-infrared wavelengths)-would be particularly valuable. Such a comparison would help to better contextualize the efficacy of the proposed approach and clarify its potential advantages and limitations.

Reviewer comments

Abstract:

- The scientific names of all microorganisms should be presented in italics throughout the abstract.
- The abstract should explicitly report the light dose (fluence) delivered after each irradiation protocol, as this information is essential for understanding and reproducing the experimental conditions.
- The sample size should be presented (n).

Introduction

- The following statement appears to be more appropriate for the Discussion section rather than the Introduction, as it anticipates and interprets efficacy outcomes:
"Nevertheless, researchers continue to assess their fungicidal efficacy through clinical trials and in vitro studies. However, the results are conflicting. Systematic reviews of Nd:YAG laser monotherapy have reported both promising and poor onychomycosis cure rates. When the review was limited to randomized controlled trials, the results showed limited efficacy.
- The Introduction would benefit from a more detailed description of the two fungal species evaluated, including their clinical relevance, biological differences, and known susceptibility to laser-based therapies.
- The authors should expand the discussion on the mechanism of action of the Nd:YAG laser, including photothermal effects, depth of penetration, and the safety and efficacy of laser application up to specific energy doses.
- A brief but clear overview of antimicrobial photodynamic therapy (a PDT) should be included, particularly to contextualize the present approach in relation to conventional antimicrobial phototherapies and also to the PDT with a photosensitizer agent.

Methodology:

- Essential experimental details are missing and should be clarified:
 - o What was the sample size (n)?
 - o How many irradiation sessions were performed?
 - o How many biological and technical replicates were included?
 - o How many occasions the experiment was performed?
- References supporting the biofilm formation protocol and microorganism cultivation methods should be provided.
- Section 2.3 should clearly report the light dose (J/cm²) delivered during irradiation.
- It is unclear whether serial dilutions were performed prior to plating for CFU determination.
- Microscopic analysis: The described methodology is appropriate and coherent for morphological (SEM) and structural/functional (CLSM) analyses of fungal biofilms. However, essential experimental details are lacking, which compromises reproducibility, methodological clarity, and scientific robustness.
 - o The authors should specify the image acquisition parameters and the number of images analyzed per experimental condition.
 - o The optical parameters used for fungal autofluorescence detection in CLSM (excitation and emission wavelengths) should be clearly detailed.

Results:

- Figure 1: Why was the same graphical scale not used for both fungal strains, as this choice limits direct comparison?
- CFU results: Although a statistically significant reduction was observed, the reduction did not reach 1 log. Therefore, despite statistical significance, the magnitude of the effect appears limited and may not be clinically relevant.
- In Figure 2, the results for both microorganisms should be expressed in Log.
- Section 3.2 should more clearly explain the format and rationale for CFU data presentation.
- The results should include the percentage of reduction for XTT (3.3) and CV (3.1), as well as the reduction in log CFU (3.2).
Lines 295-302 (3.5): The figure is incorrectly cited as Figure 5.2. This should be corrected to Figure 6.

Discussion:

- While the study addresses a relevant topic and employs established methodologies, substantial revisions are required to improve clarity, reproducibility, and the interpretation of biological relevance, particularly regarding the magnitude of the antimicrobial effect observed.
- The manuscript reports a "14-log reduction in *Candida albicans* viability". This value appears biologically implausible and requires careful verification. The authors should review the literature.

Conclusion:

There is a significant gap in the manuscript; without a formal conclusion, the implications of the XTT, CV, and log CFU data are not clearly established or supported.

Reviewer #2 (Comments for the Author):

The manuscript describes the evaluation of how 1064-nm Neodymium-Doped Yttrium Aluminum Garnet (Nd:YAG) laser affects biofilm traits of two fungal species, *Trichophyton rubrum* and *Candida albicans*, in vitro. One strain of each species was cultured to form biofilms, which were then exposed to the laser for distinct exposure times. After these biofilms were evaluated for biomass, culturable population (colony-forming units - CFU), metabolism, and microscopy. The topic is of importance in the field; however, there are some points for revision, as follows:

- Line 192: It is unclear what "up to 7 d" means, considering that plates were incubated for 24h. Please clarify.
- Lines 213-219: As written, it appears that only *T. rubrum* was assessed via confocal. Thus, it is important to provide a rationale for not using confocal for *C. albicans*. Also, based on the images displayed in Figure 5, was it Differential Interference Contrast (DIC) confocal microscopy? The reason is that no fluorophore was used, and there is autofluorescence from *T. rubrum*, but the data-collection settings are not detailed. Please provide more details or a reference for the methodology.
- Lines 221-225: Based on the description, it appears that the experiment per strain was done once, with 4 replicates, which may not be representative. Please provide a rationale for not performing the experiments at least three times to verify data reproducibility. Thus, as written, the study reads like a pilot study with preliminary data. Hence, the discussion and conclusion should be toned down.

Minor:

- Abstract: Line 51: It is necessary to define ECM.
- Species names cited in the document need revision.

Overall, the English language requires careful revision, as the manuscript does not consistently employ appropriate scientific and academic terminology.

In general, the study presents methodologies that are applicable and consistent with the proposed experimental design. In particular, the methods employed to assess the outcomes are robust, and the experimental procedures are adequately described. However, the magnitude of the reductions reported throughout the manuscript appears to be overemphasized, especially in the Discussion section.

Although biofilms are well known to be complex and resilient structures that increase microbial resistance by limiting the effectiveness of therapeutic approaches (Li Y, Sun G, Xie J, Xiao S, Lin C. Antimicrobial photodynamic therapy against oral biofilm: influencing factors, mechanisms, and combined actions with other strategies. *Front Microbiol.* 2023 Jun 9;14:1192955. doi: 10.3389/fmicb.2023.1192955.), previous studies have reported more pronounced reductions in microbial viability when phototherapies are applied, particularly classical photodynamic therapy employing a photosensitizing agent. (Chen B, Sun Y, Zhang J, Chen R, Zhong X, Wu X, Zheng L, Zhao J. *In vitro* Evaluation of Photodynamic Effects Against Biofilms of Dermatophytes Involved in Onychomycosis. *Front Microbiol.* 2019 Jun 7;10:1228. doi: 10.3389/fmicb.2019.01228.). In this context, a more comprehensive discussion comparing the observed results with those obtained from conventional photodynamic therapy approaches—employing a photosensitizer in combination with a LED light source (including near-infrared wavelengths)—would be particularly valuable. Such a comparison would help to better contextualize the efficacy of the proposed approach and clarify its potential advantages and limitations.

Reviewer comments

Abstract:

- The scientific names of all microorganisms should be presented in *italics* throughout the abstract.
- The abstract should explicitly report the light dose (fluence) delivered after each irradiation protocol, as this information is essential for understanding and reproducing the experimental conditions.
- The sample size should be presented (n).

Introduction

- The following statement appears to be more appropriate for the Discussion section rather than the Introduction, as it anticipates and interprets efficacy outcomes:

“Nevertheless, researchers continue to assess their fungicidal efficacy through clinical trials and in vitro studies. However, the results are conflicting. Systematic

reviews of Nd:YAG laser monotherapy have reported both promising and poor onychomycosis cure rates. When the review was limited to randomized controlled trials, the results showed limited efficacy.

- The Introduction would benefit from a more detailed description of the two fungal species evaluated, including their clinical relevance, biological differences, and known susceptibility to laser-based therapies.
- The authors should expand the discussion on the mechanism of action of the Nd:YAG laser, including photothermal effects, depth of penetration, and the safety and efficacy of laser application up to specific energy doses.
- A brief but clear overview of antimicrobial photodynamic therapy (a PDT) should be included, particularly to contextualize the present approach in relation to conventional antimicrobial phototherapies and also to the PDT with a photosensitizer agent.

Methodology:

- Essential experimental details are missing and should be clarified:
 - What was the sample size (n)?
 - How many irradiation sessions were performed?
 - How many biological and technical replicates were included?
 - How many occasions?
- References supporting the biofilm formation protocol and microorganism cultivation methods should be provided.
- Section 2.3 should clearly report the light dose (J/cm^2) delivered during irradiation.
- It is unclear whether serial dilutions were performed prior to plating for CFU determination.
- Microscopic analysis: The described methodology is appropriate and coherent for morphological (SEM) and structural/functional (CLSM) analyses of fungal biofilms. However, essential experimental details are lacking, which compromises reproducibility, methodological clarity, and scientific robustness.
 - The authors should specify the image acquisition parameters and the number of images analyzed per experimental condition.
 - The optical parameters used for fungal autofluorescence detection in CLSM (excitation and emission wavelengths) should be clearly detailed.

Results:

- Figure 1: Why was the same graphical scale not used for both fungal strains, as this choice limits direct comparison?
- CFU results: Although a statistically significant reduction was observed, the reduction did not reach 1 log. Therefore, despite statistical significance, the magnitude of the effect appears limited and may not be clinically relevant.
- In Figure 2, the results for both microorganisms should be expressed in Log.
- Section 3.2 should more clearly explain the format and rationale for CFU data presentation.
- The results should include the percentage of reduction for XTT (3.3) and CV (3.1), as well as the reduction in log CFU (3.2).

Lines 295–302 (3.5): The figure is incorrectly cited as Figure 5.2. This should be corrected to Figure 6.

Discussion:

- While the study addresses a relevant topic and employs established methodologies, substantial revisions are required to improve clarity, reproducibility, and the interpretation of biological relevance, particularly regarding the magnitude of the antimicrobial effect observed.
- The manuscript reports a “14-log reduction in *Candida albicans* viability”. This value appears biologically implausible and requires careful verification. The authors should review the literature.

Conclusion:

There is a significant gap in the manuscript; without a formal conclusion, the implications of the XTT, CV, and log CFU data are not clearly established or supported.

March 22th, 2026

Christina Cuomo

Editor-in-Chief

Microbiology Spectrum

Dear Dr. Christina Cuomo

We would like to thank you and the reviewers of the *Microbiology Spectrum* for taking the time to review our article. We have made some corrections and clarifications in the manuscript after going over the reviewers' comments. A separate "Marked Up Manuscript file" with tracked changes from the original submission was submitted. The changes are summarized below:

Responses to editor's comments

Comment 1: You should emphasize in improving the clarity of the text and increase the technical replicates of the experiments. Also, present each replicate in each bar graph using individual symbols.

Our response: We agree with the reviewer on how increasing the number of technical replicates can strengthen the statistical robustness of the results. In the present study, a minimum of three technical replicates were performed for the assessment of biofilm biomass, viability, and metabolic activity. We believe that this number of replicates is sufficient to ensure statistical validity and is consistent with commonly accepted experimental practices in similar studies.

We also appreciate the suggestion to display individual replicate values using symbols on the bar graphs. However, we believe that presenting the data as mean \pm standard deviation allows for clearer visualization of the overall trend in biofilm inhibition across different laser exposure times and improves readability of the figures. Therefore, we retained the current graphical format in Figures 1–3.

For transparency, the individual replicate measurements are provided in table format below this response. If required, these data could be included as supplementary material; however, since they correspond directly to the values summarized in Figures 1–3, we believe their inclusion in the main manuscript may be redundant.

Table S1. OD values of crystal violet staining (OD 600 nm) after consecutive 1064 nm Nd:YAG laser exposure. For each fungal strain, four individual experiments were conducted.

Strain	Replication	Laser exposure time (s)			
		Control	10	20	60
T. rubrum	1	0.7133	0.2669	0.2545	0.1325
	2	0.5688	0.2617	0.2002	0.246
	3	0.6122	0.0556	0.2356	0.141
	4	0.6077	0.2739	0.0576	0.0545
C. albicans	1	0.7789	0.3737	0.464	0.3381
	2	0.871	0.5475	0.4464	0.353
	3	0.83	0.6326	0.4226	0.4327
	4	1.1454	0.6053	0.4403	0.3939

Table S2. Fungal colonies count (number of colony-forming droplets) of *T. rubrum* after consecutive 1064 nm Nd:YAG laser exposure. Three individual experiments were conducted.

Replication	Laser exposure time (s)			
	Control	10	20	60
1	16	14	8	7
2	16	9	8	4
3	15	11	10	5

Table S3. Fungal colonies count (CFU/ml) of *C. albicans* after consecutive 1064 nm Nd:YAG laser exposure. Four individual experiments were conducted.

Replication	Laser exposure time (s)			
	Control	10	20	60
1	1.66×10^6	1.42×10^6	1.08×10^6	0.5×10^6
2	2.3×10^6	1.24×10^6	1×10^6	0.58×10^6
3	3.2×10^6	1.2×10^6	0.4×10^6	0.8×10^6
4	3.8×10^6	1.2×10^6	0.6×10^6	0.2×10^6

Table S4. OD values of XTT assay (OD 450 nm) after consecutive 1064 nm Nd:YAG laser

exposure. For each fungal strain, five individual experiments were conducted.

Strain	Replication	Laser exposure time (s)			
		Control	10	20	60
T. rubrum	1	0.728	0.495	0.44	0.354
	2	0.716	0.455	0.396	0.448
	3	0.745	0.597	0.395	0.517
	4	0.755	0.482	0.531	0.249
	5	0.607	0.434	0.441	0.383
C. albicans	1	2.451	1.374	0.809	0.805
	2	2.328	1.06	0.625	0.512
	3	2.418	1.11	0.781	0.642
	4	2.14	1.007	0.979	0.998
	5	2.068	1.146	0.927	0.9

Responses to reviewer 1's comments

[Abstract]

Comment 1: The scientific names of all microorganisms should be presented in italics throughout the abstract.

Our response: We thank the reviewer for pointing this out. The abstract in the manuscript file already used the correct italic formatting for microorganism names; however, the formatting was lost in the submission system. We have now corrected this, and all scientific names in the abstract on the submission site are properly italicized.

Comment 2: The abstract should explicitly report the light dose (fluence) delivered after each irradiation protocol, as this information is essential for understanding and reproducing the experimental conditions.

Our response: We thank the reviewer for this suggestion. To ensure clarity and reproducibility, we have added the specific irradiation parameters—including fluence (200 mJ/cm²), pulse duration (0.1 ms), spot size (1.5 mm), and frequency (30 Hz)—to both the abstract and the Methods section. (P2 L36–37)

Comment 3: The sample size should be presented (n).

Our response: We thank the reviewer for this suggestion. Our experiments used standard strains of *T. rubrum* (NCCP 22456) and *C. albicans* (NCCP 32703) obtained from the National Culture Collection for Pathogens (NCCP) of Republic of Korea. As only one strain of each species was used, we have provided the strain numbers in the abstract and Methods section instead of a numerical sample size (n). (P2 L35–36)

[Introduction]

Comment 4: The following statement appears to be more appropriate for the Discussion section rather than the Introduction, as it anticipates and interprets efficacy outcomes: "Nevertheless, researchers continue to assess their fungicidal efficacy through clinical trials and in vitro studies. However, the results are conflicting. Systematic reviews of Nd:YAG laser monotherapy have reported both promising and poor onychomycosis cure rates. When the review was limited to randomized controlled trials, the results showed limited efficacy."

Our response: We agree that this statement, which interprets the antibiofilm efficacy of the Nd:YAG laser, is more suitable for the Discussion section. Accordingly, we have moved these sentences to the Discussion (P26 L541–547).

Comment 5: The Introduction would benefit from a more detailed description of the two fungal species evaluated, including their clinical relevance, biological differences, and known susceptibility to laser-based therapies.

Our response: We thank the reviewer for this suggestion. To address it, we have added a paragraph in the Introduction briefly describing the clinical relevance and biofilm biology of *T. rubrum* and *C. albicans*. Since the paragraph precedes the discussion of biofilm treatment and laser therapy, we did not include their susceptibility to laser-based therapies in this section (P4 L83–100).

Comment 6: The authors should expand the discussion on the mechanism of action of the Nd:YAG laser, including photothermal effects, depth of penetration, and the safety and efficacy of laser application up to specific energy doses.

Our response: We thank the reviewer for this suggestion. To address it, we have briefly described in the Introduction that ‘selective photothermolysis’ is the primary mechanism underlying the fungicidal effect of the Nd:YAG laser and that the 1064 nm wavelength allows deeper energy penetration into nail tissues. Information regarding the safety and efficacy of laser application at specific energy doses was not included in the Introduction to maintain focus and conciseness (P6 L118–124).

Comment 7: A brief but clear overview of antimicrobial photodynamic therapy (aPDT) should be included, particularly to contextualize the present approach in relation to conventional antimicrobial phototherapies and also to the PDT with a photosensitizer agent

Our response: We thank the reviewer for this suggestion. To address it, we have added a brief description of PDT as a promising biofilm-targeting strategy for onychomycosis, including its fungicidal mechanism and limitations in the Introduction (P5 L108–116).

[Methodology]

Comment 8: Essential experimental details are missing and should be clarified. What was the sample size (n)?

Our response: We thank the reviewer for this comment. The sample size was one for both *T. rubrum* (NCCP 22456) and *C. albicans* (NCCP 32703), as the study used standard strains obtained from the NCCP. In line with common practice in experimental studies using single standard strains, the strain numbers are provided rather than a numeric sample size. Therefore, we have retained the

description of the fungal strains in the Methods section (P7 L140–141).

Comment 9: How many irradiation sessions were performed?

Our response: We thank the reviewer for this comment. In the main experiment, a single irradiation session was performed for each strain, and we have clarified this in the Methods section: “the main experiment was performed once in a single irradiation session.” (P10 L199–200)

Although only one main experiment was conducted per strain, multiple preliminary experiments were performed to determine optimal laser exposure time and experimental conditions. These preliminary experiments produced consistent results, supporting the reliability and reproducibility of the data. The Methods section now also notes that preliminary experiments were conducted under identical conditions (P10 L195–198).

Comment 10: How many biological and technical replicates were included?

Our response: We thank the reviewer for this comment. As standard strains from the NCCP were used, a single biological replicate was included for both *T. rubrum* (NCCP 22456) and *C. albicans* (NCCP 32703). Multiple technical replicates were performed as follows:

- *T. rubrum*: 4 for CV staining, 3 for CFU counting, and 5 for the XTT assay
- *C. albicans*: 4 for CV staining, 4 for CFU counting, and 5 for the XTT assay

The Methods section already notes that “each assessment included at least three technical replicates” (P10 L200), and the use of standard strains clarifies that only one biological replicate was used (P7 L140–141).

Comment 11: How many occasions the experiment was performed?

Our response: We thank the reviewer for this comment. Each strain was subjected to a single irradiation session, and the experiment per strain was performed once. We

have clarified this in the Methods section (P10 L198–199).

Comment 12: References supporting the biofilm formation protocol and microorganism cultivation methods should be provided.

Our response: We thank the reviewer for this suggestion. References have been added to support the fungal inoculum preparation (P7 L143, P8 L154), biofilm formation assay (P8 L167, P9 177), crystal violet staining (P10 L204), colony counting (P11 L217), XTT assay (P12 L252), SEM imaging (P13 L262), and CSLM imaging methods (P13 L277–280).

Comment 13: Section 2.3 should clearly report the light dose (J/cm²) delivered during irradiation.

Our response: We thank the reviewer for this comment. The fluence delivered during irradiation was 200 mJ/cm² for the Nd:YAG laser. This information has been added to Section 2.3 of the Methods (P9 L189).

Comment 14: It is unclear whether serial dilutions were performed prior to plating for CFU determination.

Our response: We thank the reviewer for this comment. For *C. albicans*, the suspension was diluted 10-fold prior to CFU plating, as described in Methods Section 2.5.2: “Bottom of the irradiated well plate was scraped with a sterile spatula and seeded into a Falcon tube containing 10 mL PBS (dilution factor: 10⁻¹).” This description has been retained. For *T. rubrum*, no dilutions were performed prior to CFU plating; we have clarified in Section 2.5.1 that the fungal suspensions were prepared “without additional dilution” (P11 L235).

Comment 15: Authors should specify the image acquisition parameters and the number of images analyzed per experimental condition.

Our response: We thank the reviewer for this comment.

For SEM, the image acquisition parameters are indicated in the lower left corner of each subfigure (Fig. 4 and 6). Because the parameters were chosen to provide the sharpest images, they vary among subfigures. In the Methods section, we now clarify that “image acquisition parameters were those that produced the sharpest images, as assessed visually (P13 L267–269).” Approximately 50 SEM images were analyzed for *T. rubrum* and 40 for *C. albicans*, with at least ten images analyzed for each laser exposure time.

For CLSM, excitation and emission wavelengths have been added to the Methods section (P14 L286–287). Approximately 40 CLSM images were analyzed for *T. rubrum*, with a minimum of ten images per laser exposure group. The Methods section now explicitly states that at least ten images were analyzed per laser exposure condition (P14 290–291).

Comment 16: The optical parameters used for fungal autofluorescence detection in CLSM (excitation and emission wavelengths) should be clearly detailed.

Our response: We thank the reviewer for this comment. For *T. rubrum* autofluorescence detection, a 405 nm excitation wavelength and an emission range of 461–465 nm were used. These parameters have been added to the Methods section (P14 L286–287).

[Results]

Comment 17: (Figure 1) Why was the same graphical scale not used for both fungal strains, as this choice limits direct comparison?

Our response: We thank the reviewer for this comment. While we agree that using the same scale could facilitate direct comparison between the two fungal strains, our primary aim was to emphasize biofilm biomass inhibition according to laser exposure time rather than compare absolute biomass between *T. rubrum* and *C. albicans*. Therefore, we retained the current graphical scales, which better

illustrate the reduction in biofilm biomass over time.

Comment 18: (CFU results) Although a statistically significant reduction was observed, the reduction did not reach 1 log. Therefore, despite statistical significance, the magnitude of the effect appears limited and may not be clinically relevant.

Our response: We thank the reviewer for this comment. We have clarified in the Results section that the CFU reduction was less than 1 log for *C. albicans* and that colony counts were reduced by less than 90% for *T. rubrum*, indicating limited biofilm viability reduction (P16 L339–341). Corresponding updates have also been made in the Abstract, Results, and Discussion to reflect that the Nd:YAG laser produced only a limited reduction in biofilm viability.

Comment 19: In Figure 2, the results for both microorganisms should be expressed in Log

Our response: We thank the reviewer for this comment. While expressing CFU counts in log scale is appropriate for *C. albicans*, our study used a modified drop-plate method for *T. rubrum*, which does not allow direct representation in log scale. We have clarified the rationale for using the modified drop-plate method in Methods Section 2.5. To improve clarity, the y-axis in Figure 2a has been relabeled as “colony-forming droplet count.”

Comment 20: Section 3.2 should more clearly explain the format and rationale for CFU data presentation.

Our response: We thank the reviewer for this comment. The rationale for using the modified drop-plate method for *T. rubrum* CFU counting has been clarified in Methods Section 2.5, including the data format and reasons for adopting this approach (P10 L214–229).

Comment 21: The results should include the percentage of reduction for XTT (3.3) and CV (3.1), as well as the reduction in log CFU (3.2).

Our response: We thank the reviewer for this suggestion. Percentage reductions for XTT and CV, as well as log reductions for CFU, have been added to Results Sections 3.1 (P15 L322–323), 3.2 (P16 L337–339), and 3.3 (P17 L354–355). The percentage reductions have also been included in the abstract (P2 L44–49).

Comment 22: <Lines 295-302 (3.5)> The figure is incorrectly cited as Figure 5. This should be corrected to Figure 6.

Our response: We thank the reviewer for pointing out this error. All incorrect citations of Figure 5 have been corrected to Figure 6 (P18 L379–387).

[Discussion]

Comment 23: While the study addresses a relevant topic and employs established methodologies, substantial revisions are required to improve clarity, reproducibility, and the interpretation of biological relevance, particularly regarding the magnitude of the antimicrobial effect observed.

Our response: We thank the reviewer for this comment. We have revised the Abstract, Results, and Discussion sections to more accurately reflect the findings. The Nd:YAG laser is now described as:

- 1) Producing limited biofilm viability reductions.
- 2) Showing inferior antibiofilm efficacy compared to aPDT.
- 3) Being suitable as an adjuvant antibiofilm modality rather than a standalone treatment.

Comment 24: The manuscript reports a "14-log reduction in *Candida albicans* viability". This value appears biologically implausible and requires careful verification. The authors should review the literature.

Our response: We thank the reviewer for this comment. The sentence in question refers to: "For *C. albicans*, previous studies showed aPDT inducing 1.0–1.4 log₁₀ and 14.33 CFU reduction..." The cited study (Rodrigues ABF, Passos JCDS, Costa

MS. *Photodiagnosis Photodyn Ther.* 2023;42:103600) reported a reduction in CFU from 31.33 ± 3.7 to 17.0 ± 1.5 , and we did not report a 14-log reduction in our manuscript.

However, we agree that the sentence could be misleading. We have revised it to state that aPDT induced a “45% CFU reduction” instead of “14 CFU reduction” to improve clarity for readers (P25 L529).

[Conclusion]

Comment 25: There is a significant gap in the manuscript; without a formal conclusion, the implications of the XTT, CV, and log CFU data are not clearly established or supported

Our response: We thank the reviewer for this comment. The Conclusion has been revised to clarify the implications of our findings (P28 L572–582). We now state that

- 1) the Nd:YAG laser has potential as an adjuvant antibiofilm therapy rather than a standalone treatment.
- 2) Biofilm biomass and metabolic activity were significantly inhibited, while biofilm viability showed only limited reduction.
- 3) Although the Nd:YAG laser was not superior to PDT, its ability to penetrate hyperkeratotic nail tissue provides an advantage in treating onychomycosis biofilms

Responses to reviewer 2's comments

Comment 1: (Line 192) It is unclear what "up to 7 d" means, considering that plates were incubated for 24h. Please clarify.

Our response: We thank the reviewer for pointing this out. The wording “up to 7 d” has been removed to avoid confusion (P12 L247).

Comment 2: (Lines 213-219) As written, it appears that only *T. rubrum* was assessed via

confocal. Thus, it is important to provide a rationale for not using confocal for *C. albicans*. Also, based on the images displayed in Figure 5, was it Differential Interference Contrast (DIC) confocal microscopy? The reason is that no fluorophore was used, and there is autofluorescence from *T. rubrum*, but the data-collection settings are not detailed. Please provide more details or a reference for the methodology.

Our response: We thank the reviewer for this comment. Confocal laser scanning microscopy (CLSM) was used to assess changes in autofluorescence of *T. rubrum* biofilms after Nd:YAG irradiation. Endogenous fluorophores in the hyphae and microconidia enable label-free visualization, and fluorescence was collected at 461–465 nm, primarily detecting intracellular NAD(P)H, an endogenous cofactor involved in metabolism. Because the same *T. rubrum* suspension used for CFU counting was imaged, changes in autofluorescence reflect laser-induced alterations in biofilm structure and cellular integrity. CLSM imaging was not performed for *C. albicans* due to insufficient intrinsic autofluorescence. To clarify, we added a subsection in the Methods titled “*T. rubrum* biofilm autofluorescence assessment by CLSM,” detailing the image acquisition parameters, the significance of autofluorescence, and the rationale for excluding *C. albicans*. (P13 L275–294)

Comment 3: (Lines 221-225) Based on the description, it appears that the experiment per strain was done once, with 4 replicates, which may not be representative. Please provide a rationale for not performing the experiments at least three times to verify data reproducibility.

Our response: We thank the reviewer for this comment. In our study, each strain was tested in a single experiment with at least three technical replicates. Although the experiments were not repeated independently three times, results were consistent across technical replicates, supporting reproducibility. Additionally, preliminary trials were conducted to determine optimal laser exposure times and experimental conditions, which also produced consistent results. The rationale for not performing multiple independent experiments has been added to

Methods Section 2.3 (P9 L195–200).

Comment 4: Thus, as written, the study reads like a pilot study with preliminary data. Hence, the discussion and conclusion should be toned down.

Our response: We thank the reviewer for this comment. The Discussion and Conclusion sections have been toned down. We now describe the Nd:YAG laser as:

1. Producing limited reductions in biofilm viability.
2. Showing inferior antibiofilm efficacy compared to PDT.
3. Being suitable as an adjuvant antibiofilm agent rather than a standalone treatment.

Comment 5: (Line 51) It is necessary to define ECM.

Our response: We thank the reviewer for pointing this out. ECM has been defined in the Abstract section (P3 L50).

Comment 6: Species names cited in the document need revision

Our response: We thank the reviewer for this comment. We have reviewed the entire manuscript and corrected spelling and formatting errors in all species names.

Sincerely yours,

Min Kyung Shin

Min Kyung Shin, MD, PhD

Professor

Department of Dermatology, College of Medicine, Kyung Hee University,

Tel: (82-2) 958-8300 Fax: (82-2) 969-6538 E-mail: haddal@hanmail.net

Re: Spectrum00091-26R1 (**Antibiofilm Efficacy of 1064-nm Neodymium-Doped Yttrium Aluminum Garnet (Nd:YAG) Laser on *Trichophyton rubrum* and *Candida albicans*: in Vitro Study**)

Dear Dr. Min Kyung Shin:

Your manuscript has been accepted, and I am forwarding it to the ASM production staff for publication. Your paper will first be checked to make sure all elements meet the technical requirements. ASM staff will contact you if anything needs to be revised before copyediting and production can begin. Otherwise, you will be notified when your proofs are ready to be viewed.

Sincerely,
Luis Martinez
Editor
Microbiology Spectrum